# Patient-Specific Solutions for Cranial, Midface, and Mandible Reconstruction Following Ablative Surgery: Expert Opinion and a Consensus on the Guidelines and Workflow

**DOI:** 10.3390/cmtr18010015

**Published:** 2025-02-13

**Authors:** Majeed Rana, Daniel Buchbinder, Gregorio Sánchez Aniceto, Gerson Mast

**Affiliations:** 1Department of Oral and Maxillofacial Surgery, Heinrich-Heine-University Duesseldorf, 40225 Duesseldorf, Germany; 2Department of Otolaryngology, Head and Neck Surgery, Mount Sinai Beth Israel, New York, NY 10029, USA; daniel.buchbinder@mountsinai.org; 3Department of Maxillofacial Surgery, University Hospital 12 de Octubre, 28041 Madrid, Spain; gsaniceto@meytel.net; 4Department of Oral and Maxillofacial Surgery, Ludwig-Maximilians-University of Munich, 80539 Munich, Germany; gmast@med.uni-muenchen.de

**Keywords:** CAD/CAM technology, cranio-maxillofacial reconstruction, patient-specific implants, 3D modeling, digital workflow, segmentation

## Abstract

Reconstruction of cranio-maxillofacial defects following ablative surgeries requires a comprehensive approach that balances functional restoration with aesthetic outcomes. Advances in computer-aided design and manufacturing (CAD/CAM) technology have revolutionized this field, enabling precise preoperative planning, including 3D modeling, segmentation, and virtual resection planning. These methods allow for the production of patient-specific implants and surgical templates while facilitating the evaluation of treatment outcomes. CAD/CAM technology offers numerous benefits, such as enhanced surgical accuracy, improved aesthetic results, reduced operative times, and the possibility of single-stage resection and reconstruction. However, limitations exist, including high costs, the need for specialized expertise, and dependency on accurate imaging data. This paper provides a surgeon-centric evaluation of the advantages and limitations of CAD/CAM in cranio-maxillofacial reconstruction. The discussion encompasses the technological workflow, clinical applications, and recommendations for optimizing outcomes. Future perspectives highlight ongoing developments, such as integrating non-ionizing imaging techniques and expanding the applicability of virtual and augmented reality. By synthesizing technical advancements and clinical expertise, this review aims to establish practical guidelines for implementing CAD/CAM technology in routine surgical practice.

## 1. Introduction

Cranio-maxillofacial surgery addresses the complex challenge of restoring both functionality and aesthetics in patients who have undergone ablative procedures. These interventions, often necessary for treating malignancies, infections, or trauma, result in significant tissue deficits involving bone, soft tissue, or both. The ultimate goal of reconstruction is to achieve functional rehabilitation while preserving or restoring the patient’s natural appearance.

Technological advancements in imaging and digital workflows have profoundly transformed the field of reconstructive surgery. The integration of computer-aided design and manufacturing (CAD/CAM) has streamlined diagnosis, surgical planning, and execution, facilitating a paradigm shift toward patient-specific solutions [1]. These approaches enable precise visualization of three-dimensional (3D) anatomical structures and the seamless integration of surgical steps, from imaging and planning to monitoring treatment outcomes.

As in all medical disciplines, the success of new methods is determined by their scientific validity and clinical applicability. While objective data play a critical role, subjective clinical expertise and practical experience are equally important in evaluating innovative technologies [2,3].

This paper synthesizes current advancements in CAD/CAM technology, focusing on its role in cranio-maxillofacial reconstruction. By outlining the essential technical and clinical requirements, we aim to provide a comprehensive guideline for implementing patient-specific solutions that ensure safety, precision, and optimal outcomes for patients with complex craniofacial defects.

## 2. Workflow

Patient-specific solutions in cranio-maxillofacial reconstruction have evolved significantly over the decades, driven by advancements in imaging, modeling, and manufacturing technologies. Digital data acquisition and CAD/CAM processes have transformed traditional workflows, enabling more precise diagnostics, personalized implant design, and streamlined surgical execution.

Importantly, modern technology complements rather than replaces the clinical expertise and principles established in cranio-maxillofacial surgery. Accurate diagnosis remains the cornerstone of successful reconstruction, supported by thorough clinical examination and high-resolution imaging. CAD/CAM workflows integrate these foundational steps with 3D modeling, virtual surgical planning, and implant design, offering a cohesive approach from preoperative planning to postoperative monitoring.

The requirements for producing implants vary by anatomical region, reflecting differences in functional demands and material properties. Each region—whether cranial, midface, or mandibular [4]—presents unique challenges that demand tailored, patient-specific solutions to optimize both function and aesthetics.

### 2.1. Clinical Examination

Accurate reconstruction begins with a detailed understanding of the patient’s clinical condition. Digital data processing cannot replace the need for surgical expertise; thus, delegating all digital steps to non-clinical staff should be approached with caution. Critical assessments include the extent of the defect, tissue quality, and anticipated functional limitations, alongside the patient’s overall health status, which can significantly influence surgical outcomes.

To streamline this process, checklists integrated into clinical workplace systems (CWS) can ensure comprehensive data collection and support surgical planning.

### 2.2. Preoperative Imaging

#### 2.2.1. Optimizing Data for Surgical Planning

Preoperative imaging is a critical component of cranio-maxillofacial reconstruction, serving to document pathological processes and assess disease progression. Common imaging modalities include X-ray, computed tomography (CT), cone beam computed tomography (CBCT), magnetic resonance imaging (MRI), scintigraphy, and positron emission tomography (PET), as well as hybrid methods like PET/CT and PET/MRI. Admission protocols for these procedures are typically determined by radiologists.

In addition to diagnostic purposes, imaging plays a key role in surgical and reconstruction planning. This may include data from intraoral, model, or surface scans, which are often specified by surgeons to meet the requirements of implant or template production.

To minimize unnecessary imaging, reduce radiation exposure, and control costs, surgeons and radiologists should coordinate to ensure that diagnostic data can also support surgical planning. Imaging protocols should be optimized for dual use, where possible, and additional data acquisition, such as intraoral or surface scans, should be planned collaboratively. Whenever possible, surgical planning should be based on recent data, preferably not older than six months; however, exceptions may be necessary depending on the clinical context. This is particularly crucial in secondary reconstructions or cases involving planned adjuvant therapies.

#### 2.2.2. Technical Considerations

CT and CBCT data are the primary imaging modalities for planning, generating 3D models, and producing surgical templates or implants. While data from non-ionizing imaging methods, such as MRI or sonography, are not yet fully integrated into CAD/CAM workflows, emerging technologies like “black-bone” MRI show promise for future applications. Additionally, scan logs provide valuable data for specific applications, such as dental rehabilitation, epithesis production, or the evaluation of head and neck surface symmetry.

CT and CBCT imaging protocols typically require high-resolution imaging with axial sections at a gantry inclination of 0 degrees and a layer thickness not exceeding 1 mm. If the data are also intended for surgical navigation, bone- or tooth-supported reference markers—such as transcutaneous metal screws or removable dental splints with fiducial markers—must be attached.

High-resolution imaging is essential for these workflows. CT and CBCT protocols typically require axial sections with a gantry inclination of 0 degrees and slice thickness ≤ 1 mm. For surgical navigation, reference markers such as transcutaneous screws or removable dental splints with fiducial markers must be incorporated during imaging.

The choice between CT and CBCT depends on the clinical requirements. For cases requiring only bone visualization—such as planning reconstruction plates for bony defects—either modality is appropriate. However, when soft tissue structures, such as tumors, blood vessels, or muscles, are involved, contrast-enhanced CT is often superior to CBCT due to its better visualization of vascularized structures and a wider field of view. This is particularly critical for complex reconstructions involving vascularized grafts. For example, the suitability of fibula grafts can be assessed in terms of volume, shape, and blood supply, reducing the risk of vascular compromise in dependent regions.

In summary, high-resolution CT remains the primary imaging modality for surgical planning, implant design, and template production. CBCT can be employed in cases where soft tissue visualization is unnecessary. Additionally, scan data from dental and surface reconstructions can supplement imaging workflows, particularly in applications such as epithesis production or symmetry assessments.

### 2.3. 3D Modeling

The 3D modeling of data acquired from preoperative imaging is a prerequisite for virtual surgical planning. This process can be conducted either “in house” by surgeons or technicians, provided they have access to suitable software, or outsourced via external CAD/CAM modeling services to medical engineers with a clinical background.

#### 2.3.1. Requirements for Planning Software

Virtual planning software must meet several critical requirements to facilitate accurate and efficient workflows. The most important feature is the capability to import external datasets from diverse sources, such as CT, CBCT, MRI, intraoral scans, model scans, and surface scans. This requires compatibility with various file formats, including Wavefront OBJ, Stereo-Lithography Interface (STL), Digital Imaging and Communications in Medicine (DICOM), and American Standard Code for Information Interchange (ASCII). Among these, DICOM serves as a key standard for storing and transmitting medical images, enabling seamless integration of imaging data.

Another essential feature is the ability to combine and compare datasets for 3D congruence. This integration allows users to represent and analyze both soft tissue and bone structures within a unified 3D workspace, providing a comprehensive view of the patient’s anatomy. Emerging technologies, such as virtual, augmented, and mixed reality, further enhance planning by introducing advanced visualization and interaction capabilities.

Additional requirements for planning software include the following:

Multiplanar Views: These enable detailed visualization of the patient’s anatomy from various angles, facilitating precise surgical planning.

Osteotomy Simulation: This feature allows users to predefine surgical steps, design osteotomy templates, and contour bony donor sites.

Segmentation Functions: Segmentation isolates specific anatomical structures, such as bones, soft tissues, and nerves, reducing surgical risks like nerve damage.

A robust 3D planning software should incorporate all these features to support precise surgical planning and execution, ultimately improving patient outcomes and surgical accuracy.

#### 2.3.2. Modeling and Segmentation

The 3D planning software generates virtual anatomical models from imaging datasets, such as DICOM file sets. These models can be exported and printed in full scale if required, providing a tangible representation of the patient’s anatomy for preoperative analysis.

Segmentation is a critical step in surgical planning, as it isolates relevant anatomical structures for further analysis and modeling. This process can be performed manually, semi-automatically, or automatically. Manual segmentation, while precise, is time-intensive and highly user-dependent. Automatic segmentation methods, on the other hand, offer rapid and accurate results, with the flexibility to make manual adjustments when necessary. In reconstructive surgery, preparatory segmentation often includes separating key areas such as the upper and lower jaw, the cranium and midface, or pathological regions from surrounding healthy tissues.

Automatic segmentation relies on various algorithms or their combinations, such as threshold-based methods and 3D interpolation of regions of interest. Some advanced software incorporates anatomical unit-based region augmentation algorithms to further enhance segmentation accuracy. Depending on the specific planning software, different solutions are tailored to meet clinical and technical needs.

For patient-specific implants (PSI), segmentation quality is paramount. Essential structures to be segmented include the bone and its anatomical subunits, such as the mandible, maxilla, and orbit. Other critical elements include the resected area, preserved anatomical features such as teeth or the mandibular canal with the inferior alveolar nerve, and any pre-existing osteosynthesis hardware, such as plates or screws.

In outsourced CAD/CAM modeling services, surgeons must meticulously review and approve all segmented structures to ensure accuracy before advancing to the next stages of surgical planning.

Pearl: A critical step in planning is the repositioning of bony segments into their correct anatomical position. This involves aligning the proximal segment with the temporomandibular joint in the glenoid fossa, expanding a collapsed dental arch, restoring pre-existing occlusion, and addressing displaced pathological fractures, infected osteoradionecrosis, or continuity defects in secondary reconstructions. Failure to properly reposition these segments can lead to irreversible planning errors and jeopardize surgical outcomes.

### 2.4. Virtual Surgery Planning

After completing the modeling and segmentation process, virtual 3D planning becomes the next step in cranio-maxillofacial reconstruction. For primary reconstructions, the process begins with the virtual simulation of resection and reconstruction, allowing surgeons to define surgical margins and design reconstruction plans. In secondary reconstructions, planning starts by analyzing data on the current treatment status and addressing anatomical and functional challenges based on prior surgeries or complications [5,6].

#### 2.4.1. Planning Mode

Virtual planning can be approached in three distinct modes, depending on the resources, expertise, and clinical requirements.

##### In-House Planning by the Surgeon

In-house planning requires the surgeon to complete all technical steps independently, including the segmentation of anatomical structures, the design of resection templates, and the virtual placement of implants. While this approach is more time-consuming, it enables the surgeon to execute and review individual surgical steps virtually, which helps in anticipating technical difficulties or risks.

Once the virtual planning is complete, the surgeon can use the collected data to print stencils or 3D models, which are subsequently sent to the device manufacturer for validation and production. These outputs may include surgical guides, physical 3D models, and patient-specific implants such as plates [7].

##### Service Model

The service model is a more collaborative and convenient approach for the surgeon. In this model, planning data and information are shared between the surgeon and a clinical engineer via an online platform. The engineer operates the planning software under the surgeon’s guidance, implementing adjustments as required. This collaborative workflow allows the surgeon to focus on clinical decision making while relying on the engineer for technical execution.

Once a treatment option is finalized, the surgeon reviews the plans, provides written consent, and assumes responsibility for the surgical outcomes. This includes approving the design and production of patient-specific templates and implants.

Pitfall: It is critical that the operating surgeon is actively involved in the planning sessions to ensure that the virtual plan aligns with intraoperative realities and personal surgical techniques.

##### Backward Planning

Backward planning is a specialized approach often used for osseointegrated dental implants or prosthetic anchors. This method ensures that reconstruction facilitates optimal functional and aesthetic outcomes by considering the desired postoperative state first.

For example, in dental implant planning, the existing occlusion or a virtually designed postoperative occlusion serves as a guide. This ensures that the planned bone graft’s position and volume allow for the precise insertion of dental implants [8].

#### 2.4.2. Virtual Planning for Resection

Virtual planning for the removal of pathological findings shares similar criteria and challenges with traditional clinical and radiological planning. Current clinical standard planning software is primarily designed for bony resections, where determining resection margins and ensuring safety distances—particularly when surrounding soft tissue is involved—relies on clinical and radiological findings. This responsibility ultimately falls on the planning surgeon.

One of the key advantages of virtual planning is its 3D visualization capability, which allows for detailed visualization from multiple spatial directions and the overlay of image data. However, achieving definitive resection safety remains beyond the capabilities of virtual planning alone. The most reliable confirmation of resection status still depends on histopathological analysis of resection specimens. Nevertheless, advancements in medical imaging and diagnostics hold promise for improving this process in the future.

When planning a resection, several fundamental considerations must be addressed. First, the extent to which the resection can be planned for the specific pathology should be evaluated. Pathologies confined to bone are often easier and safer to remove compared to combined bone and soft tissue pathologies. Similarly, anterior findings are typically easier to delineate than posterior ones. Based on this assessment, and considering the patient’s overall health, the necessary reconstructive steps should be planned. Comprehensive reconstruction with bony restoration as a primary therapy may be possible without a final histopathological report. Alternatively, a two-stage approach can be taken, with virtual planning in the primary therapy limited to creating resection templates and a patient-specific reconstruction plate.

Another key consideration is whether enlarging safety margins could reduce the risk of incomplete resection. Surgeons must critically weigh the potential consequences of such an approach: Would it result in the unnecessary loss of healthy, functionally or aesthetically important tissue? What impact would a larger defect have if the planned reconstruction failed? These questions are central to balancing oncological safety with optimal functional and aesthetic outcomes.

The surgical approach must also align with resection planning. It is important to determine the anticipated surgical overview, working angles, and risks to surrounding structures. Depending on the case, a transoral, transfacial, or combined approach may be necessary for a safe resection. This decision influences the production and use of resection templates, which will be addressed in Section 2.5.1. If bony reconstruction is planned at a later stage, the resulting defect configuration should be carefully coordinated. This includes considering the direction of insertion for the bone graft, the selected soft tissue access, and the configuration of contact zones between local bone and graft during planning. Secondary reconstructions may require additional bone resections to optimize defect configuration, refine bone stumps, and prepare for graft placement.

As described, resection planning involves defining spatial resection boundaries, selecting appropriate soft tissue access, and planning the reconstruction procedure. The resection boundaries are segmented within the 3D model of the planning software, enabling visualization and protection of functionally important structures such as teeth, nerves, or the tear duct. After segmentation, the resected specimen can be virtually “removed” to evaluate the resulting defect configuration and make necessary adjustments. Once the desired resection extent and defect configuration are finalized, measurements of size and volume can be performed to document the process and plan subsequent steps.

To implement the plan clinically, resection templates (cutting guides) are usually created, as detailed in Section 2.5.1. Alternatively, a navigated approach without templates may be employed, provided that appropriate preparations are made.

#### 2.4.3. Planning the Reconstruction

Reconstruction planning is influenced by numerous factors. Similar to conventional planning, considerations must extend beyond the local defect to include the patient’s overall health, prior treatments, the condition of potential tissue donor sites, and the disease prognosis, including potential further treatments. These factors give rise to different reconstruction scenarios, each requiring tailored strategies.

Complete primary reconstruction (patients with and without adjuvant therapy who will undergo complete soft tissue and bony reconstruction as part of the ablative procedure)

For patients undergoing complete primary reconstruction, involving both soft tissue and bone reconstruction as part of the ablative procedure, virtual 3D planning offers significant advantages. Combining soft tissue and bone grafts is essential, and 3D planning facilitates graft selection by incorporating preoperative angio-CT imaging of potential donor regions (leg, hip, or shoulder). These datasets, modeled and segmented in planning software, allow for evaluating vascular supply, bone volume, and graft shape for optimal compatibility with the defect area.

Virtual placement of transplants into the defect region further enhances precision. For lateral defects, the intact opposite side can be mirrored and segmented to guide transplant adjustments. For central defects, standard models may be used, followed by virtual fitting of bone grafts. Key anatomical landmarks such as the foramen magnum, clivus, or crista galli help ensure symmetrical alignment of the 3D skull model.

Pearl: Restoring the original jaw shape is not always ideal. For edentulous patients with mandibular prognathism, optimizing dental rehabilitation may involve reducing the chin arch relative to the upper jaw.

To achieve symmetry and enable future dental rehabilitation, transplants may require virtual osteotomy and contouring. Compared to conventional planning, 3D planning provides several advantages:Determining whether opening or wedge osteotomies are feasible based on vascular supply and transplant type.Calculating precise bone segment sizes and positions to avoid nonunion and ensure adequate vascularization.Minimizing donor site morbidity by predefining bone length and shape, especially in the scapula and iliac crest.aProtecting perforators in combined soft tissue and bone transplants.Ensuring proper orientation of the vascular pedicle for easy connection to local vessels.

Pitfalls: In combined bone-soft tissue grafts, the vascular pedicle may have a limited reach compared to single grafts. Additionally, bone segment lengths below 2–3 cm can compromise vascularization. Special attention must be paid to soft tissue coverage, especially in upper jaw reconstructions with vascularized grafts. Excessive graft volume at the alveolar process level may compromise blood circulation due to compression by the lower jaw.

After the virtual bony reconstruction has been completed, the next step is to plan the alloplastic fixation. The decision about the osteosynthesis material and positioning depends on the location of the defect (cranium, midface, lower jaw), the graft type, and experience-based surgical preferences.

Partial primary reconstruction (patients with and without adjuvant therapy to be reconstructed after resection with soft tissue and alloplastic bone replacement)

In cases where reconstruction is limited to soft tissue and alloplastic bone replacement after resection, 3D planning remains invaluable. It provides data for STL models, drilling templates, and reconstruction plate production, facilitating precise surgical preparation. Each case must be carefully evaluated to ensure optimal outcomes.

Late primary or early secondary bony reconstruction (patients who are scheduled to undergo rapid bony reconstruction after receiving the resection results)

Patients undergoing rapid bony reconstruction after resection results benefit from patient-specific implants (PSI), particularly in dentate patients with mandibular defects requiring occlusion preservation. PSIs offer precise fits and guided screw hole positions, eliminating the need for intraoperative plate bending. Preoperative adaptation of prefabricated reconstruction plates using STL models can also save time.

Pearl: When pathological infiltration prevents implant bending at the mandibular edge, virtual planning enables anatomical reconstruction and ensures resection safety margins. This approach also allows the creation of a specific implant, avoiding intraoperative complications.

Whether the implant can be reused for bony reconstruction depends on histopathological findings. Plate dimensions should consider potential re-resections, though cost remains a limiting factor for PSIs compared to prefabricated options.

Secondary bone reconstruction (patients for whom bony reconstruction is only planned at intervals, possibly also after adjuvant therapy)

Patients undergoing staged bone reconstruction typically exhibit significant scarring and shrinkage of the soft tissue surrounding the defect area, which can lead to complications such as inward or outward plate extrusions. This can be improved by choosing adapted implant configurations. In the cranial and midfacial area, freehand bending of plates or meshes is usually not a problem. In the lower jaw area, however, bending reconstruction plates requires practice and inaccuracies are difficult to avoid. Suitable instruments are scarce, especially for plate configurations with angle formation over the edge or torsions, and “atypical” bending increases the risk of material breakage in assembled plates.

In these cases, virtual reconstruction planning is very useful. Each plate configuration can be planned on individual 3D models and subsequently created using CAD/CAM processes. Lingual or caudal plate positions, rounded corners, reduced circumferences, etc. can help in preventing extrusions in jaw stumps [9]. Nevertheless, basic rules must be observed, which are mostly based on experience with prefabricated panels (see Section 2.5). The use of these implants for later bony reconstruction is rarely possible, as it would require new planning to account for the tissue changes that may have occurred in the meantime.

Patients Not Scheduled for Bone Reconstruction

In general, similar principles apply as in the previous section. The aim is to provide bony support and stabilize movable bone stumps.

In the cranium and midfacial area, implants should be covered with soft tissue, and direct contact with the sinuses, the nasal cavity, and exentered orbits should be avoided. Exposed material is prone to contamination, leading to infections and further loss of soft tissue cover, especially in irradiated patients.

In the lower jaw area, the use of alloplastic materials for permanent bone replacement is still critical and cannot be predicted. Edentulism may be advantageous in this situation. Common complications such as material extrusions, plate and screw fractures, avulsions from the bone, and fractures are also observed with patient-specific implants. Nevertheless, virtual planning can facilitate an easier adaptation of the plate configuration to the soft tissue conditions. These include reducing the chin arch, individually determining the screw positions for a more widespread distribution of forces, and rounding plate surfaces to protect soft tissue, thereby enhancing the likelihood of success.

Secondary reconstruction (patients both with and without adjuvant therapy, who should only be reconstructed in a second step following resection)

This affects a minority of patients who, for various reasons, did not undergo reconstruction following resection. Additionally, there are patients whose reconstruction failed due to complications, or those requiring epithetic tissue replacement after resection. In these cases, a careful analysis of the current situation is imperative before contemplating reconstructive options.

In contrast to primary reconstructions, in secondary reconstructions the tissue changes that have already occurred must be considered and corrected if necessary. In the cranial and midfacial areas, this usually involves the lack of soft tissue volume when bony reconstruction is required. In addition, important structures such as eyes or the external nose may be missing. In the lower jaw, aside from soft tissue problems, there may also be misalignment of the jaw stumps, temporomandibular joint problems, and loss of occlusion with and without loss of mobility. Especially after irradiation, mobility can be completely restricted, making anatomical reduction challenging and physiological movements impossible. To address these issues, additional measures such as coronoidectomy may be necessary.

In these cases, an initial virtual planning helps with analysis.

Cranium and midface

After resection, it is crucial to identify and validate both soft tissue and bony volume deficits. To achieve this, data from CT scans of the skin surface or surface scans are imported into the planning software and overlaid onto bone data. Depending on the selected display mode within the 3D model, users can visualize the skin surface alone, the skin surface combined with transparent bone structures, or only the bone.

Through mirroring or overlaying standard models, the volume required to achieve symmetry can be accurately assessed. This enables the creation of detailed planning models for tissue transfers by exporting and printing STL files. The choice between isolated soft tissue transfer, soft tissue transfer with an alloplastic carrier, or combined bone–soft tissue transfer becomes more precise and objective when based on 3D visualization rather than clinical assessment alone. Once the type of transplant has been selected, harvesting templates can be designed for both soft tissue and bone.

For vascularized transplants, virtual planning offers the distinct advantage of accurately positioning the vascular pedicle in relation to the transplant’s shape. Additionally, the production of patient-specific alloplastic supports, such as meshes, scaffolds, or plates, can be tailored to protect functional structures like the brain or eyes while supporting soft tissue. This approach is particularly valuable for complex three-dimensional shapes. By eliminating manual bending and cutting processes, patient-specific implants offer smoother edges, rounded contours, and precise fixation points in areas with adequate bone volume, ensuring better fit and function.

Another option is the virtual planning of support plates for epitheses and obturators designed to replace anatomical structures such as the nose, orbits, or tooth-bearing regions. By comparing the available bone supply and optimizing the orientation of base plates, even complex cases can be addressed with patient-specific implants. Furthermore, the volume requirements for epitheses and obturators can be estimated during planning. In cases where space is limited due to prior interventions, preoperative tissue thinning can be virtually planned to ensure optimal fit and function.

Mandible

Reconstruction of the mandible begins with assessing the position of the jaw stumps and correcting their alignment if necessary. After symmetrical alignment of the virtual 3D model and segmentation of the jaw stumps, anatomical adjustments are made based on the condyle position and occlusion. In cases where the condyle is missing on one side, mirroring techniques can compensate for asymmetry and guide further planning. Although occlusion can be set virtually, this process is time-consuming in current planning software and often hampered by inaccurate CT imaging due to metal artifacts. To overcome this limitation, dental scans of the remaining teeth or jaw impressions in occlusion can be performed. The resulting “occlusion data” are integrated into the planning software and serve as the basis for precise jaw positioning.

For edentulous patients, alignment relies on anatomical landmarks, such as the center of the chin, or by superimposing standard models. Once the jaw stumps are virtually adjusted to the desired position, bony reconstruction, soft tissue reconstruction (if required), and the design of alloplastic fixation implants follow the principles outlined in Section 2.4.3, “Complete Primary Reconstruction”. If occlusion adjustments are needed, creating occlusal splints as in orthognathic surgery is recommended. These acrylic splints, “printed” to the desired thickness based on the planning data, assist in segment adjustment and later verify the achieved positioning.

#### 2.4.4. Dental Rehabilitation

Dental rehabilitation is an integral part of mandible reconstruction planning, especially in patients requiring implant-supported dentures following ablative surgery. For bony or combined bone–soft tissue reconstructions, dental considerations should be incorporated early in the planning process. The position and volume of the reconstructed bone must allow for subsequent dental implantation.

While primary dental implantation during bony reconstruction is feasible in selected cases, it is more commonly performed as a secondary measure. In primary reconstruction, dental implants can be pre-inserted into the donor bone to heal and condition the site before transplantation. After healing, the donor bone with implants is transferred to the recipient site. Virtual 3D planning plays a crucial role in this complex process, ensuring accurate spatial orientation of the bone and implants between donor and recipient areas. Drilling and sawing templates, along with prepared osteosynthesis plates, further support this procedure. This approach is generally reserved for cases without time constraints and where the pathological condition is clearly understood.

Secondary dental implantation, now a standard practice in implantology, requires highly skilled surgeons in the context of jaw reconstructions. Ensuring sufficient bony volume to accommodate implants and precise positioning for prosthetic treatments is critical. During virtual reconstruction planning, bony requirements should take precedence, as soft tissue adjustments can often be made secondarily. Planning software integrates dental implant dummies, enabling surgeons to virtually assess occlusal requirements, avoid interference with osteosynthesis material, and address unsuitable positions in osteotomy gaps. Following bony reconstruction, updated imaging data can validate the outcomes and guide further dental prosthetic care. This workflow supports a fully digital, interdisciplinary approach based on backward planning principles.

##### Final Review and Documentation

Before final approval and production, the treatment plan must be critically reviewed by the surgeon. Each planning step should be documented, with key views captured as screenshots to ensure transparency. These planning documents can be integrated into the operating room if technology allows and used for training purposes, such as educating medical students, trainee doctors, and for case discussions with colleagues. Additionally, the documentation holds significant forensic value. If visualizing the plan on a monitor or printout is insufficient, STL models can be created for each planning step.

Pearl: In bony reconstructions, creating separate STL models for the defect and the adapted bone grafts has proven successful. After sterilization using X-rays, these models allow surgeons to intraoperatively verify the defect’s correspondence with the plan, ensure the graft fits precisely, and confirm the positioning of osteosynthesis plates. This approach is particularly valuable for vascularized transplants and upper jaw reconstructions, where virtual planning may not fully account for the surrounding soft tissue layer.

### 2.5. Guides and Implants Design

#### 2.5.1. Patient-Specific Resection and Drilling Guides

The primary role of patient-specific resection and drilling guides is to accurately translate virtual planning into surgical practice. To achieve this, several key requirements must be met. Table 1 shows the comparison of benefits and risks of various implant types (Table 1):
Tissue compatibility and sterilizability: Guides are commonly made from materials like titanium, polyamide, or their combinations.Identification: Guides should be marked with a registration number, usage side, and, if applicable, color-coded for easy identification.Contamination prevention: During procedures involving saws, chisels, or piezoelectric devices, guides must not shed material chips into the wound.Cooling channels: Resection or drilling guides should include irrigation channels or holes to facilitate cooling during surgery.Perfect fit and precise positioning: Accurate placement depends on artifact-free imaging and anatomically distinct structures as reference points.Secure fixation: Guides must remain fixed in position, typically achieved through preplanned screw placements.Delicate yet stable design: Guides must balance fragility and robustness to ensure safe operation, even in restricted surgical areas.Angle considerations: Working angles dictated by soft tissue access must be factored into the guide design, particularly for sawing, drilling, or piezoelectric procedures.Drilling templates: These must include correctly sized drill channels and angles, with optional sleeves for dental implant placement.Technical support features: Resection templates should be equipped with flanges or slots to ensure precise cuts.Protection of vascularized transplants: Guides should safeguard the vascular pedicle and perforators in transplant procedures.Easy removal: Guides must be simple to remove after use.Cost efficiency: Manufacturing costs should remain low to make these guides widely accessible.

Pitfall: Achieving precise positioning of resection templates can be challenging when anatomical features like recesses or protrusions are sparse. This often tempts planners to design large templates, requiring extensive surgical access for placement. Subperiosteal positioning of such templates can denude healthy bone over large areas, increasing the risk of healing complications, especially in patients receiving adjuvant radiation therapy. This can lead to conditions like infected osteoradionecrosis. For this reason, templates should be designed as small as possible to balance precision with minimal tissue disruption.

##### Technical Aspects of Resection and Drilling Guides

The resection process is guided by one of two methods:
Flange Guidance: A flange on the template defines the planned resection line by serving as a physical boundary.Slot Guidance: The resection is performed within a pre-designed slot in the template.

The choice between these methods depends on surgical preferences and the specific requirements of the case.

If virtual planning encompasses both resection and reconstruction steps, combined resection and drilling guides can be employed. These guides integrate features for both processes, allowing the template to account for reconstruction plates or frameworks. Screw hole positions are pre-encoded based on computer planning, with integrated tubes accommodating standard drill sleeves for precise pre-drilling.

Such guides enable the following:
Exact plate positioning via predefined screw holes.Maintenance of correct angulation to the bone surface.Controlled depth for bicortical fixation, ensuring structural stability.Avoidance of damage to critical structures like the bulbus oculi, tooth roots, or nerves during drilling.

By precisely predefining the plate’s position via the guides, mandibular resection can proceed without prior plate fixation on the unresected mandible. This approach enhances surgical precision and efficiency while reducing potential complications.

##### Applications in Donor Sites

The use of sawing and drilling guides includes not only the resection area on the head but also the donor sites for bone transplants. The donor sites in the fibula, iliac crest, and scapula have special characteristics that will be briefly outlined.

##### Fibula

Virtual 3D-planned sawing and drilling guides have significantly improved fibula flap harvesting, offering several advantages over conventional stencils:
Angio-CT as a foundation: The data provide detailed information on blood circulation in the legs, enabling the selection of the optimal donor side and planning alternatives in case of vascular anomalies.Precise measurements and vascular assessment: Virtual planning allows exact measurement of the fibula’s dimensions, visualization of its vascular supply, and identification of necessary perforators for fasciocutaneous soft tissue portions. It also ensures preservation of critical structures, such as the caudal (6–8 cm) and proximal (8 cm) fibular stumps, to maintain ankle joint stability and protect the proximal fibula-tibial joint and the lateral collateral ligament of the knee.Contour optimization through osteotomies: Wedge osteotomy can be planned with precision, ensuring that all bone segments remain in the desired spatial plane under full contact. Flanges or guide slots in the sawing templates facilitate true-to-angle osteotomies.Drill guides for segment alignment: Integration of drill guides into cutting templates ensures the precise alignment of fibula segments to the desired shape, pre-drilled for attachment to patient-specific osteosynthesis plates. Drilling is usually monocortical to preserve vascular integrity.Time efficiency and standardization: These guides save time, standardize the procedure, reduce complications, and enable selection of the optimal bone segment for reconstruction.

Pitfall: The fibula’s straight structure and flat exterior can make precise template placement challenging. Templates should be positioned carefully, compared with preoperative plans, and secured using screws to avoid misalignment (Figure 1).

##### Iliac Crest

Virtual planning templates are also valuable for harvesting iliac crest transplants, where they facilitate accurate osteotomy and graft shaping:
Angio-CT of the pelvis as a basis: The imaging provides detailed information about the vascular supply and conditions of the iliac and femoral arteries, helping to identify stenoses caused by arteriosclerosis. It also aids in selecting the optimal donor side, shape, and size of the graft.Template fixation and complex shapes: Templates are fixed with screws and simplify osteotomy for complex structures, such as the mandibular angle and ascending mandibular branch with the articular process. Templates also help determine whether the anterior iliac spine can be preserved.

Pitfall: To avoid secondary fractures of the anterior iliac spine, ensure that the bone base remains at least 3 cm wide after graft removal.

Pearl: Opening osteotomies are typically used to bend iliac crest transplants and increase graft length. Virtual planning allows for simulation of these osteotomies, conserving graft length and reducing donor site morbidity. After the osteotomy, the openings can be closed with cancellous bone removed during harvesting.

Scapula

Bone harvesting from the scapula presents unique challenges due to its complete encasement within a muscle cuff, preventing direct template placement on the bone surface. Virtual 3D planning overcomes these difficulties, providing several advantages:
Angio-CT of the shoulder region as a basis: This imaging allows visualization of the scapula’s vascular supply, aiding in the selection of the donor site, such as the lateral scapular edge (a. circumflexa scapulae) or scapular tip (a. angularis scapulae). It also highlights the residual bone configuration, helping assess the risk of secondary fractures in narrow bone bridges.Virtual transplant fitting: The scapula is virtually modeled, transferred to the defect site, and adjusted to match the recipient area’s requirements. This ensures an optimal fit for complex reconstructions, such as basal maxillary defects requiring a horseshoe-shaped graft.Template supports and vascular preservation: Virtual planning determines the stilt-like supports for sawing and drilling templates, placed outside the transplant boundaries. These supports are screwed to the underlying bone after muscle repositioning, with careful attention to preserving the vascular supply along the lateral scapular edge.Guided osteotomies: Osteotomies are performed after muscle separation, guided by the template frame. For added flexibility, split stencils can be used when needed (Figure 2).

#### 2.5.2. Patient-Specific Implants—Mandible

The three main types of reconstructive and replacement surgical implants for the mandible are patient-specific reconstruction plates, condylar replacements, and cribs/trays. Each type addresses specific functional and reconstructive requirements, as detailed below.

a.Patient-specific reconstruction (PSR) plates

Patient-specific reconstruction plates are alloplastic osteosynthesis materials designed to ensure mandibular continuity in a load-bearing and functionally stable manner, either alone or in combination with bone transplants. The process from virtual 3D planning to CAD/CAM production allows complete technical freedom in positioning and plate design. While PSR plates offer notable advantages over prefabricated options, they also entail specific disadvantages and risks.

Advantages:
Precise planning and visualization: Virtual planning ensures accurate positioning of the plate and screws, reducing the risk of damaging critical structures such as the inferior alveolar nerve, teeth, or dental implants.Optimal positioning: Factors like local bone availability, soft tissue conditions, additional interventions, and aesthetic considerations (e.g., avoiding chin over-projection) are integrated into the design.Time efficiency during surgery: Preoperative planning and the use of drilling templates eliminate the need for intraoperative plate bending.Material integrity: Plates are manufactured with a perfect fit, eliminating mechanical stress from overbending and reducing the risk of material breakage.Soft tissue compatibility: Patient-specific plates avoid unnecessary edges and corners, minimizing tissue irritation compared to prefabricated options.Improved force distribution: Plates allow free screw positioning, enhancing stability in cases of reduced bone density.Increased plate stability: Sections without required screw fixation are manufactured without screw holes, maintaining plate strength while optimizing thickness.Adaptability: PSR plates are particularly beneficial in cases with complex anatomical shapes or restricted adaptability of prefabricated plates, including controlled transoral or lingual applications.

Disadvantages:Time-intensive production: The design and manufacturing process requires significant time, making PSR plates unsuitable for emergencies.Irreversible errors: Planning mistakes, such as incorrect condyle positioning, cannot be corrected after the plate’s production.Limited flexibility: Plates configured for specific bone transplants may not accommodate alternative grafts if the initial transplant fails.Persistent complications: Issues such as plate breakage, screw loosening, and fractures near screws occur with PSR plates, though comparative data with prefabricated plates remain scarce.Cost: PSR plates are more expensive than prefabricated alternatives.

Risks:Complex designs: Virtual planning can lead to overly complex solutions, particularly when used by inexperienced individuals. These designs may not align with soft tissue conditions, increasing the risk of complications.Lack of evidence-based data: Despite decades of use, there is limited data on optimal plate size, shape, or load conditions in varying scenarios. Conventional plates are often overengineered for maximum load, while CAD/CAM PSR plates demand tailored designs that remain underregulated for technical and legal reasons.

Pitfall: To minimize legal and clinical risks, PSR plates should align closely with the configurations of conventional plates. Only professional manufacturers should produce these implants, ensuring compliance with medical device regulations. In-house production or unusual designs increase liability, and all legal considerations must be addressed prior to implantation.

Key Parameters for Plate Design:

The plate design can be described using a few parameters:

PositionThe plate usually runs along the lower edge of the lower jaw body and along the posterior edge of the ascending lower jaw branch. To optimize force distribution, the plate can be doubled in a fork shape in the ascending ramus of the lower jaw and in the chin region, according to the bone trajectories. In the defect area, the plate’s course depends on whether a bone graft should be fixed or if the defect should only be closed with soft tissue.When using bone grafts, it is advantageous to slightly move the plate cranially. For fibula transplants, the bone should be relocated more into the alveolar process area, unless doubling (overbarrel and underbarrel) is planned. This facilitates later dental implant rehabilitation. In the case of iliac crest and scapula transplants, the vascular pedicle is spared.If the defect area is only closed with soft tissue, moving the part of the plate bridging the defect lingually can reduce tissue tension over the plate and lower the risk of material exposure [9].LengthPlate length is determined by the size of the defect. When using locking screws for plate fixation, there should be space for at least three screws. For non-locking screws, at least four screws are recommended if there is sufficient residual bone volume, with a minimum distance of five millimeters from the resection edge to the first screw. If conditions permit, increasing the number of screws will not do any harm. It will create a reserve in case bony resection becomes necessary.Regardless of the defect length, in case of edentulous, atrophic lower jaws, an attempt should be made to fix the screws in the better-supplied, muscle-covered areas of the chin and jaw angles/ascending lower jaw regions.ThicknessDifferent profile heights can be used depending on the procedure. Low profiles (2 mm) can be used for bony reconstructions, while thicker plates with higher profiles (2.5 mm) are recommended for continuity reconstructions without bone grafts.Screw holes/screwsAs previously described in connection with the type of screw used, the number of screw holes in the local bone varies. It is advisable to use at least three locking screws per side and at least four non-locking screws per side. These screws are typically fixed bicortically.

Pearl: In situations of anticipated poor bone quality, e.g., in cases of severe osteoporosis, it is advisable to consider a better distribution of force between the screws. By spreading the plate ends in a fan shape, the screw positions can be changed from a linear to a triangular arrangement, thereby enhancing stability.

To secure bone grafts, at least two screws per segment should be inserted. For fibula transplants, monocortical fixation is usually sufficient, whereas for iliac crest and scapula transplants, bicortical fixation ensures more stability. However, this must be weighed against the risk of vascular pedicle injury.

Angulation and drilling vectorsTo protect functionally important structures such as nerves or teeth, drill holes can be angled up to a maximum of 15 degrees. The angulation is predetermined with the drilling templates, so only in cases of very loose bone structure it is necessary to ensure the desired angle is maintained during screwing.

b.Condylar replacement

Condylar replacements for the mandible address a range of defect sizes and can be optimized through virtual 3D planning. The following options are commonly employed, depending on the defect characteristics and clinical requirements:
Osteochondral rib grafts (costochondral transplants):Osteochondral rib grafts are indicated for short-length defects involving loss of the condyle and condylar process. Virtual resection planning enables precise design and implementation of these grafts:3D Planning and Size Determination: The size and shape of the graft are defined using CT data and translated into sawing and drilling templates for either primary or secondary reconstruction.Patient-Specific Plates: These plates stabilize the relatively soft rib bones and support graft positioning, ensuring secure fixation and functional restoration.Condylar plates or plate extensions with a condylar attachment:Condylar plates are useful when the pathological resection status is unclear or the disease prognosis is uncertain.Indications: Primarily considered as a temporary solution, condylar plates may be used intraoperatively if tumor infiltration requires resection of the condyle. In such cases, patient-specific solutions are often not feasible.Preoperative Planning: If identified preoperatively, condylar plates can be designed using 3D planning and CAD/CAM production. The condyle shape may be derived from the following:
oThe patient’s resected condyle (if unaffected by pathology).oA mirrored condyle from the opposite side.oA standardized condyle model.
Production: The manufacturing process mirrors that of reconstruction plates. Ready-made condylar add-ons can also be used with patient-specific plates for added versatility.

Pitfall: When using condylar plates or add-ons, it is crucial to monitor the bony joint socket. Improper stabilization may lead to collapse into the middle cranial fossa, resulting in complications. While patient-specific condyles may lower this risk, reliable comparative data remain scarce [10].

3.Total temporomandibular joint replacement:If temporomandibular joint replacement is planned as a permanent solution, the use of a total joint implant is recommended. The components of the condylar fossa, the condylar head, and support plates for fixation are custom-made and manufactured with different materials adapted to the necessary requirements, such as titanium, cobalt-chromium-molybdenum, and ultra-high molecular weight polyethylene. The use of “metal-on-metal temporomandibular joint prostheses” should be avoided due to evidence indicating increased wear rates. Currently, manufacturing is time-consuming and expensive, and only a few recognized manufacturers offer a fully digital solution. The surgical procedure is complex and is typically reserved for experienced surgeons to minimize the risk of serious complications. Intraoperative navigation can be employed to support position control during implantation [11].

4.Free-ending revascularized bone grafts:Free-ending revascularized bone grafts are indicated for long-distance defects with loss of the condyle, condylar process, ascending ramus of the mandible, mandibular angle region, and the corpus. They are particularly suitable for patients both before and after radiation, as well as after the loss of alloplastic endoprostheses. The grafts must be designed to ensure that a free end of the graft replaces the condylar portion and adequately supports the posterior height of the lower jaw. Virtual 3D planning, followed by production of cutting and drilling guides for resection and flap rising, as well as reconstruction plates for fixation, greatly simplifies the approach. In addition to selecting the most suitable bone donor site (fibula, iliac crest, scapula), precise planning of the shape, size, the position of the vascular pedicle, and the production of patient-specific reconstruction plates are important to ensure optimal functional and aesthetic outcomes for the patient.

Pearl: Reconstruction plates should include stable suspension points to counteract gravitational pull and tissue tension. This is especially critical in maintaining joint function and minimizing complications.

c.Crib/Tray

Despite controversial and predominantly negative discussions, the method of closing mandibular bone defects using grid-shaped supports for cancellous bone or alloplastic bone replacement material has received renewed attention, thanks to the possibility of producing patient-specific cribs [12]. Increased stability is obtained through direct connection to a reconstruction plate, an exact fit, and rounded shape and edges without the need for bending. In addition, the administration of growth factors should guarantee improved success. Future studies will need to address whether the method’s weak points, such as the risk of extrusions, inadequate blood supply to the transferred cancellous bone, or the alloplastic bone replacement material with lack of bony transformation, can be compensated for.

#### 2.5.3. Patient-Specific Implants—Cranium and Midface

Patient-specific implants for the cranium and midface address a variety of reconstructive and functional needs. The main types of implants and their characteristics are summarized in the Table 1 below.

**Table 1 cmtr-18-00015-t001:** Comparison of Benefits and Risks of Various Implant Types.

Implant Type	Use	Material	Advantages and Key Features	Risks and Considerations
Osteosynthesis Plates	Fixation of grafts/flaps in cranial and midfacial regions	Titanium, Mini-/Microplates	- Pre-planned shape and position for optimal bone contact - Integration of vascular pedicle positioning - Avoidance of interference with dental implants - Pre-drilled holes ensure precise positioning and fixation	- Requires careful planning for vascular pedicle and transplant position
Orbital Plate	Reconstruction of orbital walls	(See AO Foundation White Paper)	- Highly precise implants via 3D planning and CAD/CAM - Custom-designed for individual defect morphology	- Discussed in detail in the AO Foundation White Paper [13]
Cranioplasty Implants/Meshes	Closing cranial bone defects	Titanium, PEEK, Polyethylene	- Extremely precise, dimensionally stable implants - Sterilizable, CAD/CAM-produced - Pre-planned fixation points reduce insertion time	- In midface area, risk of exposure and infection (especially in irradiated patients) - Autogenous tissue is preferred for long-term use
Fixation Aids	Supporting epitheses, obturators, or dental prosthetics	Patient-specific Substructures	- Anchoring elements like magnets or screws - Backward planning facilitates interdisciplinary integration with prosthodontics and epithesis specialists	- Requires careful coordination with interdisciplinary teams


*Details on Specific Implant Types*


a.Osteosynthesis Plates for Graft/Flap FixationNumerous conventional plating systems are available for bony fixation in the cranial and midfacial areas. These plates are typically shaped, cut, and bent intraoperatively, which presents no technical challenges when using mini- or microplates. Screws of 4–6 mm are sufficient for most fixations.The advantage of patient-specific plates lies in their pre-planned design, which incorporates critical factors:Plate shape and position are optimized for bone contact.Vascular pedicles and supplying vessels are accounted for, especially in midface defect closures.Future dental implant positions can be anticipated and interference avoided.Pre-drilled holes in patient-specific plates allow simultaneous fixation and positioning of the graft, simplifying the procedure and reducing complications.

b.Orbital PlateThe restoration of orbital wall structures is a well-established application of patient-specific 3D planning and implant manufacturing. Detailed information on orbital plates is available in a separate AO Foundation white paper [13].

c.Cranioplasty Implants and MeshesMaterials like titanium, polyethylene, and PEEK are commonly used for cranial reconstructions. These materials are sterilizable and can be CAD/CAM-processed or 3D-printed, enabling virtual planning and precise production. Cranioplasty implants are characterized by the following (Figure 3):Exceptional dimensional stability and accuracy.Pre-planned holes with optimized screw lengths and angles for fixation.

**Figure 3 cmtr-18-00015-f003:**
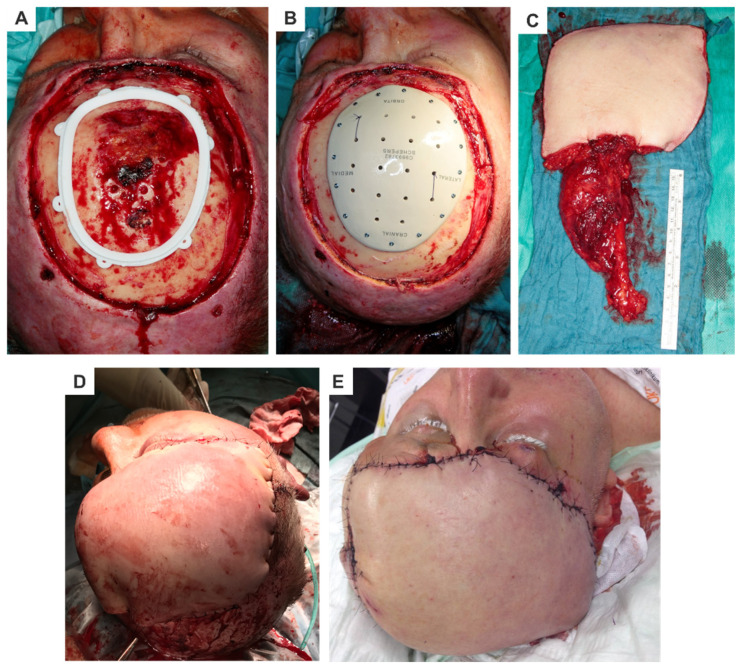
Patient-specific PEEK cranioplasty with microvascular reconstruction; (**A**) intraoperative view showing the polyamide cutting guide placed to assist in the resection of the skull; (**B**) reconstruction using a patient-specific polyether ether ketone (PEEK) osteosynthesis implant placed in the defect area, fixed with miniscrews; (**C**) harvested latissimus dorsi flap with a long pedicle before microvascular anastomosis; (**D**) immediate postoperative result after reconstruction; (**E**) second postoperative day showing the vascularized transplant.

However, challenges remain for midface applications, especially in areas around the oral and nasal cavities or paranasal sinuses. These regions are prone to material exposure, increasing the risk of infections, particularly in irradiated patients. Whenever possible, long-term reconstructions in the midface should prioritize autogenous tissue over alloplastic implants, which are better suited for temporary use [14].

d.Fixation Aids for Epitheses, Obturators, and Dental Prosthetics

Exogenous materials for esthetic and functional rehabilitation often require stable fixation. Substructures attached to the bone serve as anchoring points, using elements like magnets or screws. Virtual 3D planning ensures precise positioning and integration with prosthodontic designs. To achieve optimal outcomes, interdisciplinary planning involving epithesis specialists and prosthodontists is essential. Backward planning principles allow integration of multiple datasets from various software platforms, enabling seamless coordination.

Success in cranium and midface reconstructions relies on precise planning and execution of patient-specific implants. Good quality and complete soft tissue coverage remain essential to minimize complications and achieve long-term functional and esthetic results.

#### 2.5.4. Prototyping and Manufacturing

The production of patient-specific implants, templates, and 3D models is a critical step following analysis and CAD planning. The process begins with data exchange, where planning data are exported (commonly as STL files) and transmitted to a medical manufacturing engineer. After thorough verification for completeness and quality, the data are imported into the CAM (Computer-Aided Manufacturing) process.

Patient-specific implants can be produced using subtractive or additive manufacturing methods, each with distinct advantages and applications.

##### Subtractive Manufacturing (SM) Processes

Subtractive manufacturing involves removing material to create the desired workpiece through milling, grinding, turning, or drilling. Computerized Numerical Control (CNC) machines automate this process with high precision, allowing for complex geometries to be fabricated.

Applications: SM processes are primarily used in metal processing for producing titanium osteosynthesis plates and frameworks.Advantages: High precision and compatibility with standard industrial production.Challenges: High material consumption, as removed material cannot be reused directly. Prefabricated implants benefit from material-saving SM optimization, but similar efficiencies are not yet fully realized for patient-specific implants.

##### Additive Manufacturing (AM) Processes

Additive manufacturing involves layering material to create the workpiece based on a virtual 3D model. This method enables the production of intricate geometries with high precision. The layer thickness is critical for achieving the desired accuracy [15] (Ziegener 2020). Depending on the material, different processes are used [6].

Techniques in Additive Manufacturing [Table 2]

Polymerization (Stereolithography—SLA):Liquid photopolymers are hardened layer by layer using a laser beam. Common methods include the following:
oTwo-Photon Polymerization (2PP);oPolymer Jetting (Thermal Jet Printing);oDigital Light Processing (DLP).Applications: SLA techniques are primarily used for STL planning models and templates in reconstructive surgery.

2.Selective Laser Sintering (SLS)/Selective Laser Melting (SLM):These powder bed processes use a laser to melt metal or plastic powders selectively, based on a CAD model. Once solidified, the object is built layer by layer. SLM is particularly versatile and can process high-strength materials like titanium alloys (e.g., Ti6Al4V). Post-processing, such as heating, further optimizes the workpiece.Applications: Production of complex, patient-specific implants from metals or plastics.

3.Fused Layer Modeling (FLM):FLM involves extrusion of thermoplastic filaments through a heated nozzle, which deposits the material in dots or layers to form the structure. Applications: Rapid prototyping for models or stencils.

4.3D Printing and Binder Jetting:A 3D printer uses a liquid binder to fuse powder layers, which are applied incrementally to form the workpiece. This method supports multi-color printing for enhanced visualization of anatomical structures like nerve canals or implants.

Pearl: The ability to print in multiple colors is ideal for creating detailed 3D models. However, the precision is generally lower compared to SLA or SLS processes.

Comparison of Manufacturing Methods is summarized in Table 2.

**Table 2 cmtr-18-00015-t002:** Comparison of Manufacturing Methods: Processes, Applications, Advantages, and Challenges.

Method	Process	Applications	Advantages	Challenges
Subtractive (SM)	Removal of material using CNC machines	Metal processing for plates and frameworks	- High precision - Compatible with industrial production	- High material waste
Additive (AM)	Layering material (e.g., SLS, SLM, SLA)	Complex geometries and patient-specific implants	- Enables intricate designs - Suitable for metals and plastics - Reduces assembly time	- Requires post-processing for optimal results
Binder Jetting (3D)	Binding powder layers with liquid adhesives	Detailed anatomical models	- Supports multi-color models for enhanced visualization	- Lower precision than SLA/SLS

##### Summary and Selection Criteria

Both additive and subtractive manufacturing processes can produce patient-specific implants, templates, and models with high precision. The choice of method depends on factors such as the following:Material Requirements: SLS/SLM are preferred for high-strength metals, while SLA suits polymers.Workpiece Complexity: Additive methods handle intricate designs, while SM is ideal for simpler geometries.Desired Properties: Strength, biocompatibility, and aesthetic considerations influence the process selection.

Currently, SLS and SLM techniques are favored for producing complex geometries or difficult-to-process materials. CNC machining is preferred for simpler designs requiring robust durability.

### 2.6. Surgery

While patient-specific implants, templates, and 3D models enhance precision and efficiency, the fundamental principles of reconstructive surgery remain unchanged. Proper preparation is essential to ensure successful integration into the surgical workflow:Planning Documents: Ensure accessibility to planning documents via monitors or printouts in the operating room.Verification of Materials: Confirm that all implants, templates, and 3D models are complete, properly assigned to the patient, and sterilized preoperatively.

Pitfall: To use 3D models within the surgical site, they should be sterilized preoperatively and packaged in sterile, transparent plastic bags. The appropriate sterilization method must be discussed with the manufacturer to avoid material deformation due to heat development during hot air or steam sterilization. Gas or *γ*-ray sterilization may be better alternatives. This also applies to plastic stencils.

The choice of the appropriate approach for ablative surgery depends on the location, size, and severity of the pathological findings, as well as the planned reconstructive measures. In cases involving malignant, lymphogenic metastasizing findings, planning must include cervical lymph node dissection. If defect closure with revascularized tissue transplants is required, the availability of suitable connecting vessels must be planned accordingly.

#### 2.6.1. Approach

The choice of surgical access in reconstructive procedures depends largely on the anatomical region, the extent and location of the pathology, and the planned reconstructive measures. Advances in virtual 3D planning and the use of patient-specific templates and implants have expanded the possibilities for achieving precise and minimally invasive approaches.

For the mandible, access is most commonly achieved through an external incision, either a partial or complete collar incision. This approach provides sufficient exposure for procedures such as lymph node dissection and the preparation of connecting vessels for revascularized tissue transplants. While the transoral approach is less frequently employed in ablative surgery, it has become increasingly viable with the aid of modern technologies. By combining virtual planning with patient-specific drilling and resection templates, as well as endoscopic support, surgeons can perform precise procedures even in areas with limited visibility. If necessary, transbuccal techniques can further enhance access and visualization in challenging anatomical regions.

Access to the midface varies depending on the size and location of the pathological findings. Larger lesions often require transfacial approaches, such as the Dieffenbach–Weber–Fergusson [16]) incision, which allows for comprehensive exposure. For less extensive pathologies, transoral approaches may suffice, often in combination with transnasal techniques such as midfacial degloving. The integration of drilling and sawing templates into the surgical workflow minimizes the extent of required access, reducing trauma while maintaining precision.

In the cranium, the coronal incision remains the standard approach, offering extensive access to the cranial vault and surrounding structures. This approach is often combined with techniques for addressing soft tissue defects, such as the use of rotation or advancement flaps for plastic reconstruction. When utilizing patient-specific implants, meticulous planning of the surgical access is essential to ensure proper soft tissue closure and to preserve the vascular supply. These measures are critical to minimizing the risk of implant exposure and associated complications.

#### 2.6.2. Recipient Vessels

It is recommended to assess potential recipient vessels before performing a resection, especially if revascularized tissue transfer is planned. If problems arise, such as insufficient caliber size, fibrosis, or atherosclerosis, previously discussed alternative plans may need to be implemented. This could mean being unable to use patient-specific templates for flap rising and implants specifically tailored to the shape of the transplant.

#### 2.6.3. Resection

Currently, the use of resection (cutting) guides is primarily designed for bony findings. Therefore, especially in the case of rapidly growing tumors, it is important to compare the current clinical situation with the preoperative planning and examine soft tissue resection margins for tumor infiltrates using intraoperative frozen sections. Intraosseous structures, e.g., the inferior alveolar nerve and the soft parts of cancellous bone, can also be examined for tumor infiltrates using frozen sections, enhancing the safety of the resection [17]. The use of resection guides require precise placement. It is recommended to cross-check the positioning with the preoperative planning and, if available, with the 3D model. The target position is secured by screwing at designated positions. When a combination of resection and drilling guide is used, holes for later screw placement are drilled under adequate cooling with physiological saline solution. The resection is then performed, guided through flanges or slots using a saw, chisel or piezoelectric instrument. In cases where simultaneous lymph node dissection and tumor resection are planned in the lower jaw area, en bloc resections should be attempted.

#### 2.6.4. Reconstruction

Reconstruction methods vary by anatomical region and whether bone is involved. The Table 3 summarizes the techniques, materials, and key considerations.

Patient-specific 3D planning enhances precision and reduces risks across reconstruction methods, particularly for complex cases. The DCIA flap, scapula flap, and fibula are versatile options for bone reconstruction, with the choice dependent on defect location and patient-specific factors.

### 2.7. Accuracy Assessment

With the digital database of preoperative planning, exact comparisons of outcomes can be conducted following postoperative data collection through imaging. This approach allows the evaluation of the entire process, from planning to implementation, with a level of quality previously unattainable. Clinical studies show that plans can be executed with a high level of precision. However, they also show that achieving 100% accuracy is rare [20]. This inaccuracy can stem from technical factors, surgical procedures, performance variability, and unexpected changes during the process. Another current concern is the radiation exposure to patients caused by imaging checks using computer tomography. Identifying the underlying causes of these inaccuracies will improve the application of this technology across all stages.

## 3. Evaluation

The following Table 4 summarizes the key advantages, challenges, and future directions in patient-specific surgical planning, highlighting its impact on precision, efficiency, and interdisciplinary collaboration while addressing current limitations and areas for innovation.

## 4. Limitations

Patient-specific solutions for craniofacial reconstruction, while groundbreaking, face several limitations that constrain their universal adoption and efficacy. The primary challenge is the high cost of these technologies, encompassing software, hardware, and the production of patient-specific implants and templates. These financial barriers often limit their availability in resource-constrained healthcare systems and prevent routine use in standard clinical settings.

Another critical limitation is the time-intensive nature of planning and production processes. This restricts their use in emergency situations or urgent cases requiring immediate intervention. Furthermore, while digital tools excel in bone reconstruction, their application to soft tissue remains underdeveloped. Current systems lack robust capabilities for realistic soft tissue modeling and integration, which impairs their utility for complex defects involving both bone and soft tissue.

Radiation exposure during the frequent imaging required for planning and postoperative evaluations is another concern. Although advances in low-dose imaging hold promise, this remains a significant consideration for patients undergoing multiple interventions. Additionally, the reliance on external providers for manufacturing patient-specific solutions introduces logistical challenges, potential delays, and legal uncertainties regarding data handling and ownership.

Finally, there is a learning curve associated with adopting these technologies. Younger surgeons may have a strong grasp of digital tools but lack the comprehensive surgical experience necessary for complex reconstructions. Conversely, experienced surgeons may struggle to adapt to new workflows, emphasizing the need for interdisciplinary collaboration and targeted training programs.

## 5. Conclusions

Patient-specific technologies are transforming craniofacial reconstruction by enabling unparalleled precision and customization, particularly in bony reconstructions. Their strengths lie in their ability to integrate diagnostic data, simulate surgical outcomes, and streamline complex procedures. For secondary reconstructions and planned primary procedures in oncology, these tools provide significant benefits by improving outcomes and reducing intraoperative time. However, their time and resource demands currently limit their application in emergency scenarios.

Looking to the future, these technologies represent a transitional phase toward a more integrated and holistic approach to reconstructive surgery. Advancements in augmented reality (AR) and artificial intelligence (AI) are poised to enhance the precision and scope of patient-specific solutions, potentially bridging the gap between bone and soft tissue reconstructions. Moreover, as cost-effective manufacturing processes evolve and imaging protocols improve, broader accessibility and lower radiation exposure are likely to follow.

Ultimately, the successful application of patient-specific solutions depends on the expertise of the surgical team. Achieving optimal outcomes requires a balance between traditional surgical experience, particularly in soft tissue handling, and the technical proficiency needed to harness modern digital tools. This synergy underscores the need for continuous education and training to fully realize the potential of these groundbreaking innovations.

## Figures and Tables

**Figure 1 cmtr-18-00015-f001:**
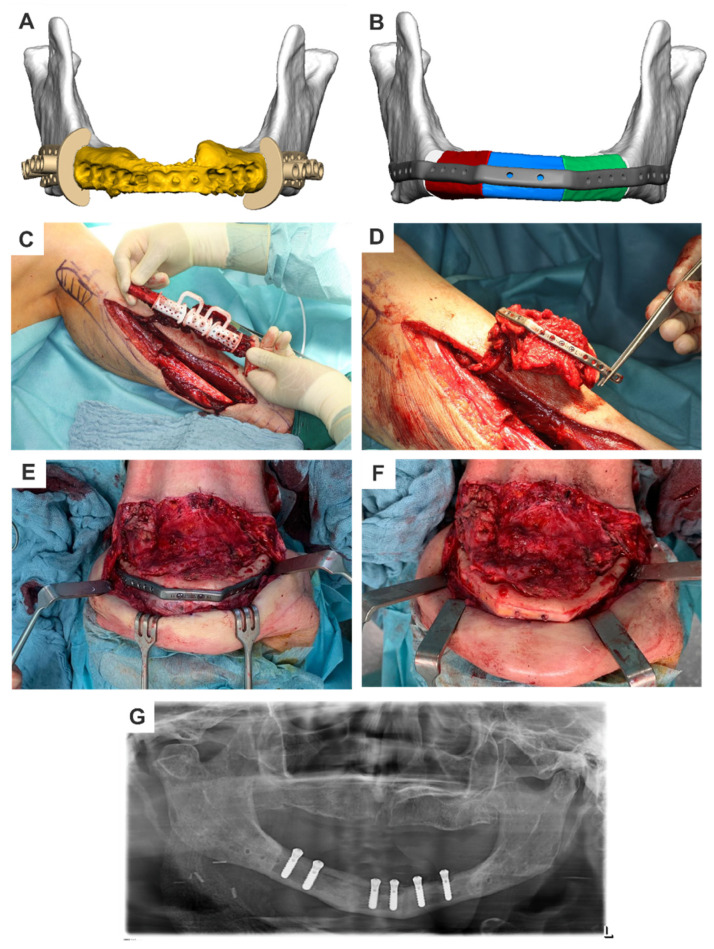
CAD/CAM-workflow for fibula flap reconstruction; (**A**) 3D model of a mandible after primary tumor resection. The cutting and drilling guides are placed at the mandible defect (yellow); (**B**) defect is virtually removed from the original mandible, the fibula segments (green, blue and red) are virtually osteotomized, wedged and bent to fit into the defect, and the patient-specific plates (PSI) virtually designed; (**C**) donor site for fibula grafting is determined based on the angiography, using polyamide cutting and drilling guides. The positioning of the cutting guide should preserve as much as possible the cutaneous vessel proximal to the osteotomy; (**D**) patient-specific implant with an osteocutaneous fibula transplant still pedicled, before fibula graft harvesting; (**E**) intraoperative image of the modified neomandible with the PSI in the definitive position; (**F**) neomandible 6 months after reconstruction and removal of PSI; (**G**) x-ray image of the neomandible with six dental implants.

**Figure 2 cmtr-18-00015-f002:**
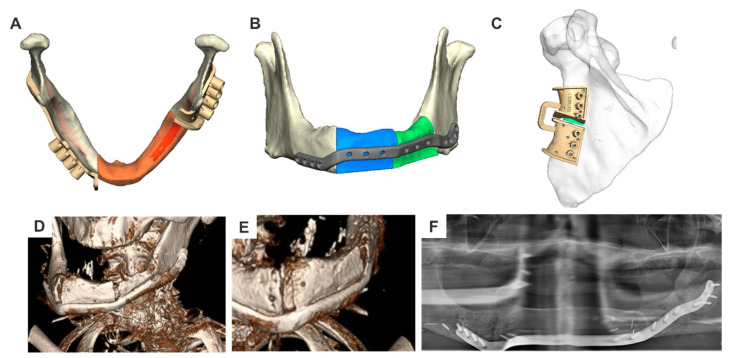
CAD/CAM-workflow for scapular free flap reconstruction; (**A**) 3D model of a mandible after primary tumor resection. The cutting and drilling guides are placed at the mandible defect (orange); (**B**) defect is virtually removed from the original mandible, the scapula segments (green and blue) are virtually osteotomized, wedged and bent to fit into the defect for an accurate dental profile, and the patient-specific plates (PSI) virtually designed; (**C**) donor site for scapula harvesting with the polyamide cutting and drilling guides. The positioning of the cutting guide should preserve soft tissue as much as possible; (**D**,**E**) postoperative CBCT showing the patient-specific implant with an osteomyocutaneous scapula transplant and the neomandible with the PSI in the definitive position; (**F**) panoramic x-ray image (orthopantomogram) of a neomandible 6 months after reconstruction.

**Table 3 cmtr-18-00015-t003:** Comparison of Reconstruction Methods: Regions, Materials, Techniques, and Key Considerations.

Reconstruction Method	Region	Material/Technique	Key Considerations
Reconstruction without bone	Cranium	- Patient-specific titanium grids, titanium plates, or PEEK implants.	- Requires sufficient soft tissue coverage. - Challenging in elderly patients or after malignancy treatment.
	Midface	- Soft tissue transfer for cavity closure. - Alloplastic grids/plates to support facial contours (in cases of secure soft tissue coverage).	- Material exposure and recurrent infections, especially after radiotherapy, pose significant risks.
	Mandible	- Load-bearing reconstruction plates, preferably patient-specific implants.	- Lingual placement reduces extrusion risk. - Occlusion stabilization (e.g., MMF screws) is critical during fixation.
Reconstruction with bone	Cranium	- Free, avascular bone grafts: Calvarial split grafts with cutting guides. - Revascularized grafts: Used for resistant infections or after irradiation.	- DCIA grafts provide stable outcomes in resistant cases. - Patient-specific implants simplify fixation.
	Midface/Mandible	- Free, avascular grafts: Supported by virtual 3D planning for precise size and shape. - DCIA Flaps: High-level reconstruction requiring microsurgery.	- DCIA flaps offer robust vascular supply for extensive defects. - Patient-specific guides enhance positioning and fixation accuracy.
	Fibula	- Cutting/drilling templates assist in segmentation and alignment.	- Ensure vascular pedicle is not compressed under the plate. - Two transfer options: fix transplant intraoperatively or pre-fix on the leg before transfer. [18,19]
	DCIA Flap	- Harvesting with osteotomy guides, ensuring vascular pedicle preservation.	- Protect vascular pedicle during transfer. - Ensure sufficient width of iliac crest to prevent fractures.
	Scapula	- Requires muscular coverage and vascular supply via subscapular axis.	- Template fixation must consider supplying vessels. - Osteotomies are completed after transplant elevation to protect the vascular pedicle.

**Table 4 cmtr-18-00015-t004:** Advantages, disadvantages, and future developments in patient-specific solutions for cranial, midface and mandible reconstruction.

Category	Details
Advantages	
Visualization	- 2D/3D visualization of bone anatomy from all angles. - Virtual mirroring of healthy contralateral structures and integration of standard models.
Accessibility	- Planning software accessible from any authorized digital device with Internet connectivity. - Service model supports non-technical users via medical technicians.
Data Processing	- Advanced 3D modeling and segmentation capabilities.
Surgical Simulation	- Accurate simulation of bone resections, removals, and reconstructions through CAD planning.
Precision Manufacturing	- Data transfer for high-precision production of patient-specific implants, templates, and 3D models (CAM).
Safety and Control	- Avoidance of sensitive structures (e.g., nerves, teeth) via visualization. - Preoperative discussions of surgical challenges in realistic virtual environments.
Time Efficiency	- Intraoperative time savings.
Results Optimization	- Improved results by eliminating critical steps (e.g., plate bending). - Exact volume determination for bone grafts minimizes surgical approaches.
Interdisciplinary Use	- Facilitates interdisciplinary data exchange (e.g., backward planning, augmented reality).
Education and Documentation	- Key tool for education, training, and surgical documentation.
Disadvantages	
Costs	- High costs for software, hardware, and CAM production.
Dependencies	- Reliance on providers for planning tools and manufacturing. - Legal ambiguities in data use.
Planning Gaps	- Limited soft tissue planning capabilities, particularly for resection of soft tissue tumors.
Complexity	- Risk of hypercomplex solutions.
Time Constraints	- Time-intensive process unsuitable for emergencies or short-term cases.
Radiation Exposure	- Increased exposure during planning and control imaging.
Expertise Distribution	- Delegation to younger colleagues with IT expertise but limited clinical experience.
Future Developments	
Technological Innovation	- Integration of artificial intelligence (AI). - Advancements in augmented reality (AR) solutions.
Cost Reduction	- Development of cost-effective solutions.
Improved Data Integration	- Simplified integration with other IT systems, backward planning, and interdisciplinary tools.
Radiation-Free Solutions	- Expansion of data sources to include low-dose or radiation-free imaging options.
Data Protection	- Improved data security and privacy measures.
Outcome Tracking	- Establishment of a comprehensive outcome registry.

## Data Availability

The data is contained within this article.

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
