# Peer review of "Patient-Specific Solutions for Cranial, Midface, and Mandible Reconstruction Following Ablative Surgery: Expert Opinion and a Consensus on the Guidelines and Workflow"

_1943-3883, 2025, doi:10.3390/cmtr18010015_

Round 1
Reviewer 1 Report
Comments and Suggestions for Authors
The authors did a very comprehensive review of computer-assisted surgery in ablative and reconstructive surgery. The paper offers a lot of tricks and considerations in different clinical scenarios, which will be beneficial for surgeons with less experience in computer-assisted surgery. I would recommend publication after minor revision.
1. Title: I think this paper is an expert opinion rather than a guideline. An expert opinion is based on a single or a few surgeons’ experiences, an expert consensus will need to include a wide range of experts with representability, while a guide will need high-level evidence support. Accordingly, I suggest deleting “the Guidelines and Workflow”. That is, changing from “Expert Opinion on the Guidelines and Workflow” to “expert opinion”. The same for the introduction part.
2. Some statements are quite subjective. Such as: Page 3, “Data for surgical planning should not exceed six months”. It depends on the need of surgery. If it is cancer surgery, 6-month is way too long.
3. Some terms are confusing, such as: “Closing osteotomies” for fibula (Page 12). It is better to use “wedge osteotomy” or “closing wedge osteotomy”. P15, “overbarrel and underbarrel” fibula. It is more appropriate to say “upper barrel and lower barrel”.
Comments on the Quality of English LanguageEnglish is acceptable but can be further improved.
Author Response
Dear Reviewer,
We sincerely appreciate your valuable feedback and the time you have taken to review our manuscript. Your positive comments on the comprehensiveness of our work and its practical relevance for surgeons with less experience in computer-assisted surgery are highly encouraging. Below, we provide our responses to your thoughtful suggestions: 1. Reviewer’s Comment: The title should reflect that this paper is an expert opinion rather than a guideline. The phrase “Guidelines and Workflow” should be removed to avoid any misinterpretation.
- Our Response: We fully understand your concern regarding the distinction between expert opinion, expert consensus, and guidelines. Our intention with the phrase “Guidelines and Workflow” was not to imply the development of a formal guideline but rather to provide structured recommendations based on expert experience. Since the manuscript aims to offer a practical and structured approach to computer-assisted surgery, we believe that keeping “Workflow” in the title helps highlight its pragmatic nature. However, we are open to modifying the phrasing to ensure clarity. For instance, an alternative could be “Patient Specific Solutions for Cranial, Midface and Mandible Reconstruction following Ablative Surgery: Expert Opinion and a Consensus on the Workflow” to better reflect the content while addressing your concern.
2.Reviewer’s Comment: The statement on Page 3, “Data for surgical planning should not exceed six months,” is too rigid and may not be applicable to all cases, especially in cancer surgery.
- Our Response: We completely agree that a six-month limit should not be generalized across all cases. We have revised this statement to reflect a more flexible approach, acknowledging that the timeframe depends on the specific surgical indication.
3. Reviewer’s Comment: Some terms should be adjusted for clarity and alignment with standard surgical nomenclature:
- Page 12: “Closing osteotomies” should be replaced with “wedge osteotomy” or “closing wedge osteotomy.”
- Page 15: “Overbarrel and underbarrel” fibula should be changed to “upper barrel and lower barrel.”
- Our Response: Thank you for these precise suggestions. We have updated the terminology accordingly to ensure consistency with widely accepted surgical terms.
We sincerely appreciate your constructive feedback, which has helped us refine and clarify our work. We hope that our revisions, along with the rationale for our title choice, address your concerns. We remain open to further discussion and would be grateful for your perspective on our proposed title modification.

Reviewer 2 Report
Comments and Suggestions for Authors
This is a very well-written paper with significant importance to the changing field of AI in craniofacial surgery. It provides important points, and the flow and explanation are clear. It will definitely benefit the field and the younger generation.
Author Response
Dear Reviewer,
We sincerely appreciate your kind and encouraging feedback on our manuscript. It is truly gratifying to hear that you recognize the significance of our work in the evolving field of AI in craniofacial surgery. Your acknowledgment of the clarity, structure, and potential impact of our paper—particularly for the next generation of surgeons—is highly valued.
We are grateful for your time and thoughtful evaluation, and we remain committed to advancing research in this dynamic field. Your support reinforces our belief in the importance of integrating AI-driven advancements into surgical practice to enhance patient outcomes and clinical workflows.
Thank you once again for your insightful review and positive assessment. We look forward to contributing further to this important area of research.